# Microbial Community and Metabolome Analysis of the Porcine Intestinal Damage Model Induced by the IPEC-J2 Cell Culture-Adapted Porcine Deltacoronavirus (PDCoV) Infection

**DOI:** 10.3390/microorganisms12050874

**Published:** 2024-04-27

**Authors:** Ying Shi, Benqiang Li, Jinghua Cheng, Jie Tao, Pan Tang, Jiajie Jiao, Huili Liu

**Affiliations:** 1Institute of Animal Husbandry and Veterinary Sciences, Shanghai Academy of Agricultural Sciences, Shanghai 201106, China; shiyingsunny@126.com (Y.S.);; 2Shanghai Key Laboratory of Agricultural Genetic Breeding, Shanghai 201106, China; 3Shanghai Engineering Research Center of Pig Breeding, Shanghai 201302, China

**Keywords:** porcine deltacoronavirus, intestinal damage, piglets, microbial community profiling, metabolome analysis

## Abstract

This study was conducted to elucidate the intestinal damage induced by the IPEC-J2 cell culture-passaged PDCoV. The results showed that PDCoV disrupted the intestinal structure and increased intestinal permeability, causing abnormalities in mucosal pathology. Additionally, PDCoV induced an imbalance in the intestinal flora and disturbed its stability. Microbial community profiling revealed bacterial enrichment (e.g., Proteobacteria) and reduction (e.g., Firmicutes and Bacteroidetes) in the PDCoV-inoculated piglet model. In addition, metabolomics analysis indicated that 82 named differential metabolites were successfully quantified, including 37 up-regulated and 45 down-regulated metabolites. Chenodeoxycholic acid, sphingosine, and oleanolic aldehyde levels were reduced in PDCoV-inoculated piglets, while phenylacetylglycine and geranylgeranyl-PP levels were elevated. Correlation analysis indicated a negative correlation between *Escherichia-Shigella* and choline, succinic acid, creatine, phenyllactate, and hippuric acid. Meanwhile, *Escherichia-Shigella* was positively correlated with acetylcholine, L-Glutamicacid, and N-Acetylmuramate. *Roseburia*, *Lachnospiraceae_UCG-010*, *Blautia*, and *Limosilactobacillus* were negatively and positively correlated with sphingosine, respectively. These data suggested PDCoV-inoculated piglets exhibited significant taxonomic perturbations in the gut microbiome, which may result in a significantly altered metabolomic profile.

## 1. Introduction

Porcine viral diarrhea is a widespread and recalcitrant epidemic infectious disease commonly found in large-scale pig farms, causing significant economic losses to the pig industry. The complex etiology and the mutating nature of the virus are significant contributors to the frequent occurrence of diarrhea [1]. Porcine Deltacoronavirus (PDCoV), emerging as a novel causative agent of porcine diarrhea in recent years, leads to mortality of up to 40% in infected neonatal pigs [1,2]. It was first discovered during a molecular surveillance of coronaviruses in birds and mammalian species in Hong Kong in 2012 [3]. Since then, PDCoV has been detected in numerous countries, including the United States, Canada, South Korea, Japan, China, Thailand, Laos, and Vietnam [4,5,6,7,8,9,10]. The pigs infected with PDCoV are more susceptible to mixed infections with other viruses, making clinical diagnosis and disease control more complicated [2]. Furthermore, recent studies have reported that PDCoV can infect cattle and poultry across species boundaries [11,12,13]. In 2021, a study reported the identification of PDCoV in plasma samples from three Haitian children [14]. Therefore, PDCoV possesses the ability to infect avian, mammalian, and even human hosts, leading some scholars to propose that it should be considered as the eighth coronavirus which can infect humans [15]. Given its ability to mutate continuously within hosts, PDCoV undoubtedly poses a potential public health concern and a threat of causing a pandemic.

As a member of the Coronaviridae family, PDCoV primarily targets the gastrointestinal tract [16,17]. Because of the significant physiologic and morphologic similarities to enterocytes in vivo, IPEC-J2 cells have been used increasingly to characterize the interactions of intestinal epithelial cells with enteric bacteria and viruses in vitro [18]. Therefore, PDCoV was firstly adapted to IPEC-J2 cells for further investigation of the intestinal damage in piglets in the study.

The intestinal barrier is categorized based on its properties into three types: the physical barrier (also known as the mechanical barrier), the biological barrier, and the immune barrier [19,20]. Any damage to these barriers can facilitate the entry of bacteria or viruses into the host’s body, enabling their passage into the bloodstream and ultimately leading to systemic damage [21]. The intestinal tract serves as a complex ecosystem, housing a diverse community of microorganisms crucial for nutrient absorption, immune regulation, and overall host health [22]. The intestinal microbiota interacts closely with the intestinal physical and mucosal immune barrier, serving as a source of microbial antigens [23,24]. The number of non-redundant genes identified in the porcine gut has reached 7.7 million [25]. The composition of the community of microbes in the porcine gut may change with nutritional status, age, environmental conditions, and health status. The intestinal environment predilection of PDCoV raises questions about its potential impact on the composition and functionality of the intestinal microbiota. Disruptions to this microbial balance can have cascading effects on the host’s metabolic processes, potentially compromising nutrient utilization and immune homeostasis. To date, no metabolomic analysis of PDCoV has been reported. Understanding the intricate interplay between PDCoV infection, piglet susceptibility, and the consequential alterations in intestinal microbiota and metabolism is imperative for devising effective preventive and therapeutic strategies. Therefore, we conducted an analysis focusing on intestinal damage at the organismal level, exploring changes in the intestinal microbiota, and mining information associated with the viral infection from the microbial community and metabolomic profiles.

This study aims to unravel the intricacies through a comprehensive analysis of 16S rRNA and metabolomics, providing insights that contribute to the development of targeted interventions against PDCoV-induced intestinal damage in piglets. The investigation revealed insights into the alterations induced by PDCoV in the piglet’s intestinal system, providing insights into the metabolic pathways and pathogenic mechanisms of PDCoV.

## 2. Materials and Methods

### 2.1. Ethics Statement

All animal experiments followed the recommendations of the Animal Care and Use Committee of the Shanghai Academy of Agricultural Science (Approval Code: SAASPZ0523075). The committee has conducted a thorough review of the protocols, encompassing operational specifics as well as measures aimed at alleviating animal discomfort and euthanasia methods.

### 2.2. Virus, Cell and Animal

The PDCoV virus (passage 6, nominated as PD-F6) was serially passaged in Intestinal Epithelial Swine (IPEC-J2) cells [26] supplemented with trypsin (10 μg/mL) in the cell culture medium. The viral titer used in this study was 10^5.5^ TCID_50_/mL. The 5-day-old piglets were acquired from a PDCoV- and PEDV-free swine herd in the Pudong district, Shanghai. All piglets were ensured to have access to sufficient milk powder.

### 2.3. Experimental Infection of Piglets with the IPEC-J2 Cell Culture-Passaged PDCoV

Ten 5-day-old piglets were randomly divided into two groups: the experimental group (*n* = 5; PDCoV-inoculated) and the mock group (*n* = 5; DMEM-inoculated). Piglets were inoculated orally with 5 mL per pig of PD-F6 [10^5.5^ TCID_50_/mL] or mock inoculated with equal DMEM. After PDCoV inoculation, the pigs were monitored for symptoms every day and then euthanized for pathological examination at PID 4. The intestine samples were fixed in 10% buffered formalin, processed, and then stained with hematoxylin and eosin (H&E) for histopathologic examination. The intestines were also snap-frozen in liquid nitrogen and then stored at −80 °C until use. The digesta in the middle intestine were selected, collected in a sterile tube, and then frozen in liquid nitrogen until microbial analysis. 

### 2.4. Microbial Community Profiling of the 16S rRNA Gene

16S rRNA sequencing of the digesta collected from the middle intestines of piglets was performed to analyze the composition of the microbial community. An E.Z.N.A.^®^ soil DNA Kit (Omega Bio-tek, Norcross, GA, USA) was used to extract the microbial DNA according to the manufacturer’s protocols. The primers 338F (5′-ACTCCTACGGGAGGCAGCAG-3′) and 806R (5′-GGACTACHVGGGTWTCTAAT-3′) were designed to amplify the hypervariable V3-V4 regions of the bacterial 16S rRNA gene. Purified amplicons were pooled in equimolar and paired-end sequenced on the Illumina MiSeq PE300 platform (Illumina, San Diego, CA, USA) according to the standard protocols by Honsunbio Technology Co., Ltd. (Shanghai, China). 

### 2.5. Metabolomic Signatures

Metabolomics analysis of stool samples was based on an LC-MS metabolomics method. Samples of 2 mL were taken in an EP tube and 600 μL MeOH (Containing 2-Amino-3-(2-chloro-phenyl)-propionic acid (4 ppm) was added followed by vortexing for 30 s. The EP tube was placed in a tissue grinder for 120 s at 50 H after steel balls were added. After ultrasound was conducted for 10 min at room temperature, the samples were centrifuged for 10 min at 12,000 rpm and 4 °C. The supernatant was filtered through a 0.22 μm membrane and then transferred into the detection bottle for LC-MS detection. The LC analysis was performed on a Vanquish UHPLC System (Thermo Fisher Scientific, Waltham, MA, USA). 

### 2.6. Bioinformatics Analysis

Sequencing reads were demultiplexed, quality-controlled by fastp (version 0.21.0), and merged by FLASH (version 1.2.7). Briefly, reads with adaptor sequences and low-quality bases (quality score < Q20) were trimmed. Truncated reads shorter than 50 bp and reads containing ambiguous nucleotides were discarded. Subsequently, the paired-end reads were merged for downstream analyses, according to the minimum overlap of 10 bp with the maximum mismatch ratio of 0.2 in the overlapping region. After removing the identified chimeric sequences, the sequences with a 97% similarity cutoff were clustered using the UPARSE algorithm. Subsequently, RDP Classifier against the reference database SILVA138 with a minimum confidence score of 0.7 was utilized to assign the taxonomy of each OTU representative sequence. Rarefaction was performed in order to compare the abundance of OTUs across samples. QIIME2 (version 2022.8) was used to import the demultiplexed sequences. Random selection of subsets of sequences was conducted to normalize the number of sequences from each sample to the lowest number of read counts.

### 2.7. Statistical Analysis

The general statistical analysis and visualized results were conducted based on R software. The Sobs, ACE, Chao1, Shannon, and Simpson indices were used to estimate the alpha diversity. To assess the differences in beta diversity between groups, principal coordinates analysis (PCoA) was conducted, which was based on Bray–Curtis matrices with statistical significance determined by permutational multivariate analysis of variance (PERMANOVA). For comparing the relative abundance of different taxa between groups, a linear discriminant analysis (LDA) effect size (LEfSe) method was performed with a *p*-value < 0.05 for the Kruskal–Wallis test and a size-effect threshold of 2.0 on the logarithmic LDA score. Spearman’s rank correlation analysis was used for correlation analysis.

## 3. Results

### 3.1. The Pathogenicity of IPEC-J2 Cell Culture-Passaged PDCoV in Piglets

The intestinal cell culture-adaptive PDCoV, PD-F6, was orally inoculated into 5-day-old piglets to clarify the enteropathogenicity. Piglets were monitored daily after PDCoV infection. Two piglets (numbered as 2 and 5) exhibited watery diarrhea at PID 1, while two other piglets (numbered as 3 and 4) developed diarrhea at PID 1.5, which persisted throughout the experiment (PID 4) with a visible reduction in the diet. One piglet (No. 1) recovered after a brief anorexia at PID 1, whereas the control piglets did not show any clinical signs during the experiment period. After autopsy at PID 4, the PDCoV-inoculated piglets exhibited gross lesions characterized as congestive intestinal walls (Figure 1A). No gross lesions were evident in the other organs of the PDCoV-inoculated piglets and negative control piglets (Figure 1B). Histologically, the ileum showed diffuse, severe villous atrophy (Figure 1C) in all the inoculated specimens except for piglet No. 1. No histologic lesions were evident in the large intestine and other organs of the PDCoV-inoculated piglets and the negative control (Figure 1D).

### 3.2. Microbial Community Profiling Analysis

Microbial community analysis was performed to analyze the difference of the microbial structure in the gut microbiota between the PDCoV-infected and control group, as shown in Figure 2. Significant differences existed in microbiota composition between the two groups. The results showed the PDCoV-inoculated group (A1) had a distinct microbiota composition that clustered differently from the control group (A2). Moreover, the microbial community structures at the phylum level of the PDCoV-inoculated group and control group are shown in Figure 2A. The abundance of each sample is presented as the percentage of the total number of sequences. PDCoV inoculation led to decreased levels of Firmicutes and Bacteroidetes, while the relative abundance of Proteobacteria was increased. Meanwhile, the ratio of Firmicutes-to-Bacteroidetes in the PDCoV-inoculated group (4.23) was higher than that of control group (1.88). At the genus level, the relative abundance of *Escherichia-Shigella*, *Limosilactobacillus*, and *Lactobacillus* was increased, while the relative abundance of *Ligilactobacillus* was decreased. The metagenome analysis LEfSe approach was applied to explore the differential gut microbiota (from phylum to genus) in the two groups of piglets (Figure 2C,D, LDA score > 2.0). Enterobacterales, Enterobacteriaceae, *Escherichia_coli*, *Escherichia-Shigella*, and Gammaproteobacteria, etc., were the most abundant differential microbiota in the PDCoV-inoculated group. In the control group, *Clostridia*, Lachnospiraceae, Lachnospirales, *Roseburia*, and *Alloprevotella* were significantly increased compared to the PDCoV-inoculated group.

### 3.3. Metabolomic Signatures of Gut Digesta from PDCoV-Inoculated Piglets

The metabolomic signatures of the intestinal digesta from PDCoV-inoculated piglets were remarkably separated from the control group in OPLS-DA score plots (R^2^X = 0.475, R^2^Y = 0.99, and Q^2^ = 0.74), suggesting a significant difference in fecal metabolites between the two groups. Significant differences between the distribution of samples were shown according to analysis of PCA (Figure 3A), OPLS-DA (Figure 3B), and model validation (Figure 3C). Furthermore, a total of 1846 differential metabolites (*p* < 0.05, VIP > 1) were identified (Figure 3D). As shown in Figure 3F, 82 named differential metabolites, including 37 up-regulated and 45 down-regulated ones, were successfully quantified. In addition, the PDCoV-inoculated group exhibited lower concentrations of chenodeoxycholic acid, sphingosine, and oleanolic aldehyde, whereas the levels of phenylacetylglycine and geranylgeranyl-PP were elevated (Figure 3E) when compared to the control group.

### 3.4. Metabolic Pathway and Network Analysis

MetaboAnalyst [27] was used to link the 82 differential metabolites to potential relevant pathways, and MetaboAnalyst 3.0 was applied to identify the impact value of relevant pathways. The results showed that five main metabolic pathways were affected, including neuroactive ligand–receptor interaction, the intestinal immune network for IgA production, the cAMP signaling pathway, alanine, aspartate, and glutamate metabolism and lysine degradation (Figure 4A). Based on the relationship between metabolomic signatures, the disturbed metabolic pathways are represented in the diagram (Figure 4B).

### 3.5. Correlation Analysis between Microbiota and Metabolomic Phenotype

The covariation of intestinal differential metabolites and the gut microbiota of genus level is presented as a heat map diagram (Appendix A) calculated by using Spearman’s correlation coefficients between microbial communities at the genus level and the 82 significantly altered metabolites. The top 20 differential metabolites with the largest abundance were extracted to draw the heat map (Figure 5) using the corrplot package of the R software. Multiple correlations between intestinal microbiota and metabolites were observed. *Escherichia-Shigella* was negatively correlated with metabolites such as choline, succinic acid, creatine, phenyllactate, and hippuric acid. Meanwhile, a positive correlation between *Escherichia-Shigella* and acetylcholine, L-Glutamic acid, and N-Acetylmuramate was identified. *Roseburia*, *Lachnospiraceae_UCG-010*, and *Blautia* and *Limosilactobacillus* were negatively and positively correlated with sphingosine, respectively. These correlation data suggest PDCoV-inoculated piglets exhibited significant taxonomic perturbations in the gut microbiome, which may result in a significantly altered metabolomic profile.

## 4. Discussion

PDCoV infects villous epithelial cells of the intestines and is enteropathogenic [2]. Cell lines derived from intestinal epitheliums recapitulating the sites of primary infection are a relevant in vitro model to study the interactions between the host and the pathogen. IPEC-J2 cells are derived from the jejunum of a neonatal piglet which was unsuckled [26], while the primary sites of PDCoV infection are located in the jejunum and ileum. IPEC-J2 cells have been reported as susceptible to infection with the propagated PDCoV [28]. Furthermore, IPEC-J2 cells have been adopted as the model cells for a pathogenecity study in several enteroviruses [29,30]. The study showed that the IPEC-J2 cell culture-adapted PDCoV (PD-F6) disrupted the intestinal structure and caused mucosal abnormalities, leading to watery diarrhea and severe enteritis.

Maintaining a homeostatic intestinal environment is crucial for the well-being of the host. Furthermore, the health of the intestine is intricately linked to the structural integrity of the intestinal epithelium, as well as the conditions within the intestinal environment [22]. The gut microbiota boasts the largest population of bacteria and the widest array of species compared to any other organ within the body. Its function extends beyond merely serving as a barricade against pathogenic organisms: it also fulfills the role of an endocrine organ by supplying vital nutrients such as short-chain fatty acids (SCFAs), vitamins, and inflammatory cytokines to the host [31]. The interactions between the structural features of gut microbiota and host metabolic phenotype in PDCoV-infected piglets have rarely been reported on until now. In the present study, integrative microbial community profiling and metabolomic signatures was employed on PDCoV-inoculated piglets compared with normal control piglets.

Reduced diversity and altered gut microbiota composition were observed in PDCoV-inoculated piglets compared with normal control piglets based on 16S rRNA gene sequencing results. This suggested that PDCoV may be linked to dynamic changes in the compositions of intestinal microbiota. The abundance of Bacteroidetes was markedly reduced in the feces of the PDCoV-infected piglets (*p* < 0.01), which aligns with previous research [32]. Bacteroidetes participate in various physiological functions, including energy provision, gut health maintenance, and the development and modulation of the immune system [33]. Furthermore, PDCoV infection seems to potentially induce dysfunction in the digestive system and other associated health concerns. Additionally, PDCoV inoculation resulted in a notable elevation in Proteobacteria levels, a diverse bacterial group that includes several pathogenic species [34]. Therefore, an increase in Proteobacteria levels in PDCoV-infected piglets could indicate a shift towards a more pathogenic gut microbiota composition. This may increase the risk of enteric infections and other related health issues in the infected animals. Moreover, Proteobacteria has been shown to interact with the immune system and modulate its response [35]. An increase in Proteobacteria levels may alter the immune balance and make the host more susceptible to immune-related disorders or infections.

Interestingly, the relative abundances of *Limosilactobacillus* and Lactobacillus increased in the intestine of PDCoV-inoculated piglets. *Limosilactobacillus* and Lactobacillus belong to lactic acid bacteria (LAB) and are considered as safe and practical probiotics that live in the intestines of humans and animals [36]. Probiotics, serving as a crucial element of the intestinal mucosal barrier, can effectively ward off the penetration of pathogenic microorganisms [37], modulate the intestinal flora [38], and produce antiviral metabolites [39]. With the in-depth study of probiotics, recent studies have explored the potential of probiotics and their related metabolites in combating intestinal viruses [40], including rotavirus (RV) [41,42], porcine transmissible gastroenteritis virus (TGEV) [43,44], and porcine epidemic diarrhea virus (PEDV) [45,46,47]. As the dominant probiotic species in the piglet intestines, the antiviral properties of lactic acid bacteria, especially those belonging to Lactobacillus, have been reported [39,40,41,42,43,44,45,46]. Currently, probiotics are widely used to prevent and treat numerous gastrointestinal disorders [48]. Therefore, the observed increase in *Limosilactobacillus* and Lactobacillus may represent the intelligent regulation of the organism’s response to repair intestinal damage and defend against further viral invasion. On one hand, PDCoV infection can disrupt the balance of intestinal flora, but, on the other hand, it triggers the activation of the body’s self-protection mechanism. Nevertheless, further exploration is necessary to fully understand the specific mechanisms and intricate relationships between PDCoV infection and the increase of Lactobacillus.

Metabolomics is a typical metabonomic research approach, which is characterized by convenient collection, non-injury, and continuity of metabolites. Fecal metabolome characterization can improve understanding of microbial responses to gut microbiota perturbations. The fecal metabolic profiles were significantly different between PDCoV-inoculated piglets and the mock ones. In PDCoV-inoculated piglets, a comprehensive analysis revealed the presence of 82 differential metabolites and five disrupted metabolic pathways. Although only two metabolites aligned with the intestinal immune network responsible for IgA production, we hypothesize that this pathway might be disrupted in these piglets. Immunoglobulin A (IgA) is the most prevalent type of immunoglobulin secreted into mucosal tissues, especially the intestinal mucus. Along with mucus and antimicrobial peptides, IgA acts as the initial barrier for intestinal epithelial cells, shielding them from various intestinal antigens. Additionally, IgA plays a pivotal role in regulating the gut microbiota (GM), including the enhancement of Proteobacteria, which are typically bound by IgA [49]. Microbial community profiling revealed an elevation in Proteobacteria, suggesting an increase in IgA levels, which aligns with the finding of the up-regulation of the intestinal immune network for the IgA production pathway, as demonstrated by the KEGG pathway analysis of fecal metabolites from PDCoV-inoculated piglets. Of course, the related metabolites of the intestinal immune network for the IgA production pathway need to be quantitatively analyzed by targeted metabolomics, and the role of this pathway in PDCoV pathogenesis should also be determined in further studies.

A significant correlation between gut microbiota and metabolites was observed, indicating that gut microbiota perturbations were closely associated with the alterations of metabolic phenotype. A marked increase in the relative proportion of *Escherichia-Shigella* was observed in the gut microbiota of PDCoV-inoculated piglets. Interestingly, a previous study indicated that expansion of Proteobacteria, the parent family of *Escherichia-Shigella*, may serve as a microbial signature of gut dysbiosis [50]. Notably, the result aligned with results presented here that expansion of *Escherichia-Shigella* was associated with gut microbiota dysbiosis. The negative correlation between *Escherichia-Shigella* and choline, succinic acid, and creatine suggested that *Escherichia-Shigella* may not utilize choline [51], succinic acid [52], or creatine [53] as a carbon or nitrogen source for growth and metabolism. Phenyllactate is a dicarboxylic acid that is produced by gut microorganisms through the fermentation of tyrosine [54]. It has been shown to have antimicrobial properties and has been used as an alternative to synthetic antibiotics [55]. A negative correlation between *Escherichia-Shigella* and phenyllactate suggested that these bacteria may not be sensitive to the antimicrobial effects of phenyllactate [56]. Hippuric acid is a conjugate of benzoic acid and glycine that is formed in the liver by the action of hippuricase. A negative correlation between *Escherichia-Shigella* and hippuric acid suggested that these bacteria may not be involved in hippuric acid metabolism or excretion. Further studies are needed to fully understand the mechanisms underlying the interactions between *Escherichia-Shigella* or other bacterium and the corresponding compounds. Understanding these interactions can provide insights into the pathogenesis of enteric infections and inform the development of novel strategies for preventing and treating these infections.

## Figures and Tables

**Figure 1 microorganisms-12-00874-f001:**
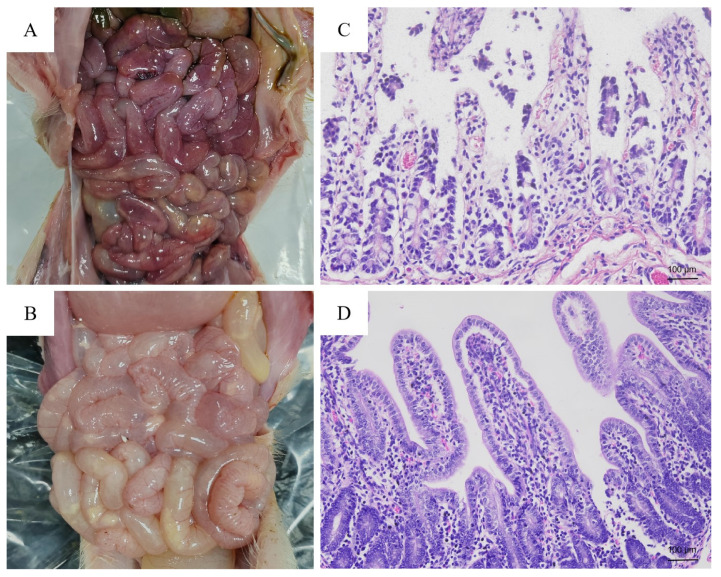
Gross and histological lesions in the small intestine of 5-day-old piglets inoculated orally with the IPEC-J2 cell culture-passaged PDCoV (PD-F6) or mock. (**A**) The gross intestinal changes in PDCoV-inoculated piglet at post-inoculation day (PID) 4. (**B**) Intestine appearance of control piglet at post-inoculation day (PID) 4. (**C**) The histological lesions in the small intestine of PDCoV-inoculated piglet by hematoxylin and eosin-staining. (**D**) The histological observation in small intestine of control piglet by hematoxylin and eosin staining.

**Figure 2 microorganisms-12-00874-f002:**
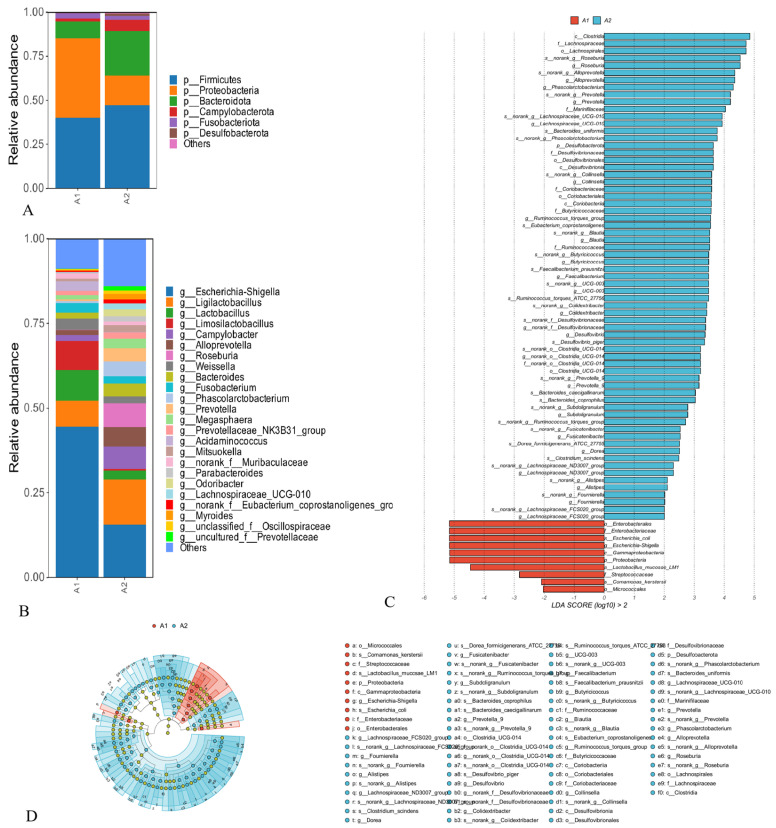
Microbial community profiling. (**A**,**B**) The community structure of intestinal bacteria in group A1 and A2 at the phylum (**A**) and genus (**B**) levels. The top species with the highest abundance for each group at each taxonomic level (phylum and genus) were selected. Different colors represent different microbes, and the ordinate indicates the relative abundance, *n* = 5. A1 (PDCoV–inoculated group) and A2 (control group). (**C**,**D**) Cladogram (**C**) and Histogram (**D**) generated from the LEfSe analysis of intestinal microbiota. *n* = 5. Only taxa of an LDA significant threshold of 2.0 were shown. A1 group (in red), taxa enrichment with a negative LDA score; A2 group (in blue), taxa enrichment with a positive LDA score.

**Figure 3 microorganisms-12-00874-f003:**
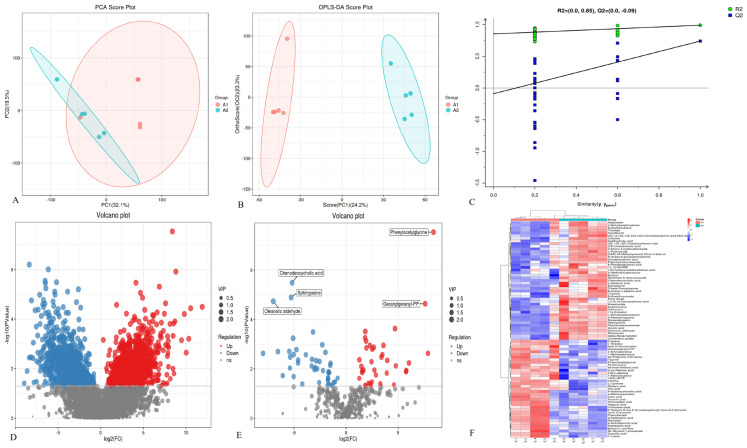
The metabolomic signatures of gut digesta from piglets. (**A**) PCA map. The distance of each coordinate point represents the degree of aggregation and dispersion between samples. (**B**) OPLS–DA map. (**C**) Model verification map of OPLS–DA. The abscissa represents the replacement retention of the replacement test; the ordinate represents the R2 (green dot) and Q2 (blue square) replacement test values; and the two dashes represent the regression lines of R2 and Q2, respectively, and positive ion mode. (**D**) Volcanic map of differential metabolites. Each point in the figure represents a specific metabolite. The abscissa represents the multiple change value; and the ordinate represents the statistical test value: that is, *p*-value. All the values are processed logarithmically. (**E**) Volcanic map of differential metabolites. (**F**) Heatmap of differential metabolites between groups. The color represents the relative abundance of the metabolite in samples.

**Figure 4 microorganisms-12-00874-f004:**
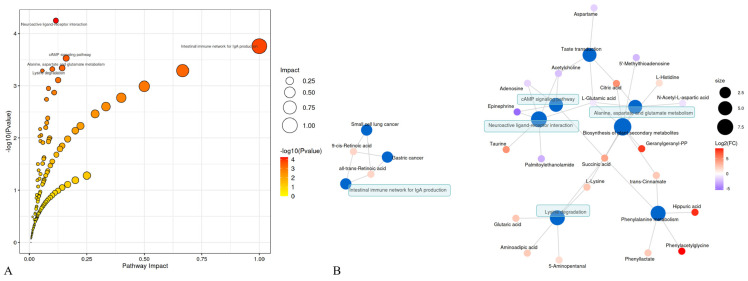
Metabolic pathways and altered intestinal metabolites in PDCoV–inoculated piglets. (**A**) Pathway analysis using MetaboAnalyst. (**B**) Metabolic pathways and altered metabolites. Blue dots indicate pathways, and other dots indicate metabolites. Red–labeled metabolites were up-regulated, and blue–labeled metabolites were down-regulated, and the darker the color, the greater the difference. Green blocks represent disturbed metabolic pathways.

**Figure 5 microorganisms-12-00874-f005:**
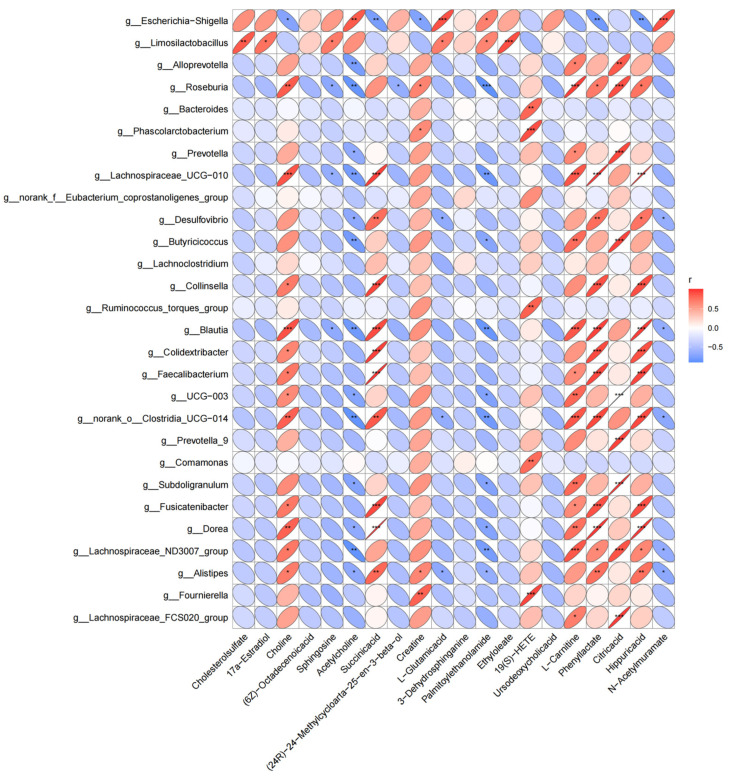
Spearman’s correlation analysis between microbiota and the top 20 differential metabolites. Positive and negative correlations are shown as red and blue in the heat map, respectively. Significant microbiota–metabolite correlations were determined based on an |r| ≥ 0.7 and *p* < 0.05 (* *p* < 0.0, ** *p* < 0.01, *** *p* < 0.001).

## Data Availability

The RNA sequencing data have been deposited in the NCBI SRA database (http://www.ncbi.nlm.nih.gov/sra, accessed on 14 February 2025), with the accession number PRJNA1071435. All data generated or analyzed during this study are included in this published article [and its Appendix A].

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
