# Peer review of "Microbial Community and Metabolome Analysis of the Porcine Intestinal Damage Model Induced by the IPEC-J2 Cell Culture-Adapted Porcine Deltacoronavirus (PDCoV) Infection"

_microorganisms, 2024, doi:10.3390/microorganisms12050874_

Round 1

Reviewer 1 Report

Comments and Suggestions for Authors

The tematic analyzed in this manuscript is of interest, but the work has deficiencies that must be corrected before publication.

The presentation of the problem in the introduction is very superficial, a more in-depth bibliographic search must be carried out (only 5 publications are referenced and the only one that is less than five years old is very general and not specifically related to the topic).

The methodological description is adequate.

In the description of the results, it should be clarified whether diffuse atrophy of the intestinal villi occurred in all inoculated specimens. Furthermore, it is strange that the authors found intestinal congestion on macroscopic examination, but did not find histopathological alterations that would confirm this observation. The remaining results are well presented.

The discussion is well presented but some aspects must be deepened, for example the changes in Limosilactobacillus and Lactobacillus found in the results.

Author Response

Thank you for your invaluable feedback. Below, I provide my response to the content that requires revision:

1. The presentation of the problem in the introduction is very superficial, a more in-depth bibliographic search must be carried out (only 5 publications are referenced and the only one that is less than five years old is very general and not specifically related to the topic).

The introduction section has been rewritten and highlighted in red.

2. In the description of the results, it should be clarified whether diffuse atrophy of the intestinal villi occurred in all inoculated specimens.

This section has been supplemented and highlighted in red in Line 164-165.

3. Furthermore, it is strange that the authors found intestinal congestion on macroscopic examination, but did not find that would confirm this observation. 

The congestion is a more intuitive manifestation in pathology, so we chose an image displaying congestion to illustrate the severe intestinal damage. Furthermore, the histopathological alterations of the intestine mainly focused on villus damage, and pathological changes related to congestion were not presented in the results. If necessary, additional pathological sections could be included.

4. The discussion is well presented but some aspects must be deepened, for example changes in Limosilactobacillus and Lactobacillus found in the results.

The discussion regarding the changes in Limosilactobacillus and Lactobacillus has been supplemented and highlighted in red in the discussion section.

Reviewer 2 Report

Comments and Suggestions for Authors

Dear Authors,

The manuscript presents very novel results and their visualization is excellent. Therefore my recommendation is published after minor revisions.

In the Ethics statement the number of the experimental procedure should be included to have the possibility of consulting it. Also in the material and methods section it should be indicated whether the diet of the animals used in the test was only breast milk or milk replacer.

Best Regards

Author Response

Thank you for your invaluable feedback. Below, I provide my response to the content that requires revision:

1. In the Ethics statement the number of the experimental procedure should be included to have the possibility of consulting it.

This section has been supplemented and highlighted in red in Line 86-89.

2. Also in the material and methods section it should be indicated whether the diet of the animals used in the test was only breast milk or milk replacer.

The diet of the animals used in the test was milk powder. It has been supplemented and highlighted in red in Line 95.